



# Atmospheric composition and climate impacts of a future hydrogen economy

Nicola J. Warwick[1,2], Alex T. Archibald[1,2], Paul T. Griffiths[1,2], James Keeble[1,2], Fiona M. O'Connor[3,4], John A. Pyle[1,2] and Keith P. Shine[5]

[1]Department of Chemistry, University of Cambridge, Cambridge, CB2 1EW, UK
[2]National Centre for Atmospheric Science (NCAS), University of Cambridge, UK
[3]Met Office Hadley Centre, Exeter, UK
[4]Department of Mathematics and Statistics, Global Systems Institute, University of Exeter, UK
[5]Department of Meteorology, University of Reading, Reading, RG6 6ET, UK

*Correspondence to*: Nicola J. Warwick (Nicola.Warwick@atm.ch.cam.ac.uk)

**Abstract.** Hydrogen is expected to play a key role in the global energy transition to net zero emissions in many scenarios. However, fugitive emissions of hydrogen into the atmosphere during its production, storage, distribution and use could reduce the climate benefit and also have implications for air quality. Here we explore the atmospheric composition and climate impacts of increases in atmospheric hydrogen abundance using the UKESM1 chemistry-climate model. We find that increases in

hydrogen result in increases in methane, tropospheric ozone and stratospheric water vapour, resulting in a positive radiative forcing. However, some of the impacts of hydrogen leakage are partially offset by potential reductions in emissions of methane, carbon monoxide, nitrogen oxides and volatile organic compounds from the consumption of fossil fuels. We derive a new methodology for determining indirect Global Warming Potentials from steady-state simulations which is applicable to both shorter-lived species and those with intermediate and longer lifetimes, such as hydrogen. Using this methodology, we

determine a 100-year Global Warming Potential for hydrogen of 12 ± 6. To maximise the benefit of hydrogen as an energy source, emissions associated with hydrogen leakage and emissions of the ozone precursor gases need to be minimised.

## 1 Introduction

The adoption of low carbon hydrogen ($H_2$) as an energy source could lead to substantial reductions in carbon dioxide emissions and a significant climate benefit. However, fugitive emissions from a hydrogen economy will increase levels of hydrogen in

the atmosphere which can further affect atmospheric composition through chemical feedbacks. It is important that the atmospheric implications of potential changes to hydrogen emissions are investigated in detail before the implementation of widespread hydrogen use.

The hydrogen abundance in the atmosphere increased over the twentieth century (Patterson et al., 2021), and today has a

mixing ratio of around 530 parts per billion (ppb), with a small inter-hemispheric gradient (±20 ppb) (Novelli et al, 1999, Patterson et al., 2021). Fossil fuel combustion and biomass burning account for approximately 50% of the current total global





hydrogen source, with the remainder arising from the oxidation of methane (CH4) and volatile organic compounds (VOCs) in the atmosphere (e.g. Ehhalt and Rohrer 2009; Pieterse et al., 2013; Grant et al., 2010). Hydrogen removal is dominated primarily by uptake to soils and reaction with hydroxyl radicals (OH). OH is the main atmospheric oxidising agent and if

hydrogen emissions were to increase, subsequent changes in the OH concentration could alter the lifetimes of important atmospheric greenhouse gases. Therefore, whilst hydrogen itself is not radiatively active, it can act as an indirect greenhouse gas.

An increase in the tropospheric concentration of hydrogen reduces the availability of OH via reaction R1. Reductions in

tropospheric OH will result in increases in the atmospheric lifetime of CH4 and its abundance (Derwent et al., 2020; Derwent and Field 2021) since the major atmospheric sink of CH4 is through reaction with OH (R2).

$$H_2 + OH \rightarrow H_2O + H \qquad\qquad (R1)$$

$$CH_4 + OH \rightarrow CH_3 + H_2O \qquad\qquad (R2)$$

The oxidation of both hydrogen and CH4 in the troposphere can lead to the generation of ozone, with impacts for both climate and air quality (e.g., Archibald et al., 2020a; Schultz et al. 2003). In the stratosphere, oxidation of hydrogen and methane will lead to increases in stratospheric water vapour with potential implications for climate and stratospheric ozone (e.g., Tromp et

al. 2003; Warwick et al., 2004). Increases in atmospheric hydrogen could also influence stratospheric ozone recovery through the production of $HO_x$ radicals (=OH+$HO_2$), which are involved in ozone destruction cycles.

The extent to which future changes in hydrogen might affect atmospheric composition and climate will depend upon the level of hydrogen leakage and the ultimate size of a future hydrogen economy. In addition, emission reductions in species currently

emitted by the production and consumption of fossil fuels, including methane, CO, $NO_x$ (=NO+$NO_2$) and Volatile Organic Compounds (VOCs) (hereafter referred to as co-emissions), will also induce feedbacks on atmospheric composition (e.g., Jacobson et al. 2008). There is significant uncertainty in the size of a future hydrogen economy and the leakage rates and co-emission reductions are uncertain, depending on the forms of new technology implemented (e.g., Frazer-Nash 2022; Lewis 2021).


Here we report on calculations with the UKESM1 (Sellar et al., 2019) chemistry-climate model on the climate and atmospheric composition effects of a range of different atmospheric hydrogen boundary conditions, consistent with a range of different scenarios for a future hydrogen economy. We determine changes in effective radiative forcing (ERF) and present an improved estimate of the hydrogen Global Warming Potential (GWP), accounting for composition changes in both the troposphere and



stratosphere. As part of our GWP calculations, we present a refined methodology for calculating indirect GWPs from steady-state simulations, appropriate for use with both short-lived and longer-lived species.

## 2 Model and scenarios

### 2.1 The UK Earth System Model (UKESM1)

The U.K. Earth System Model, UKESM1 (Sellar et al., 2019), is a state-of-science Earth system model that couples Earth
system modules to the HadGEM3-GC3.1 climate model (Kuhlbrodt et al., 2018; Williams et al., 2018). UKESM1 has a horizontal resolution of 1.875° in longitude and 1.25° in latitude, and 85 vertical levels extending from the surface to 85 km. Atmospheric composition changes are calculated using the UK Chemistry and Aerosols (UKCA) atmospheric chemistry module, which includes a coupled stratospheric-tropospheric chemistry scheme (Archibald et al., 2020b) and interactive 2-moment aerosol scheme (Mulcahy et al., 2020). UKCA includes the emissions of nine chemical species: nitric oxide (NO),
carbon monoxide (CO), formaldehyde (HCHO), ethane ($C_2H_6$), propane ($C_3H_8$), acetaldehyde (MeCHO), acetone ($Me_2CO$), isoprene ($C_5H_8$) and methanol (MeOH), while surface mixing ratios of $CH_4$, $N_2O$, CFC-11 ($CFCl_3$), CFC-12 ($CF_2Cl_2$), $CH_3Br$, $H_2$ and COS are prescribed. The photochemical sources and sinks of $H_2$ (and $CH_4$) are fully interactive, but the use of the Lower Boundary Condition (LBC) fixes the atmospheric burden of $H_2$ (and $CH_4$). LBCs are widely used in chemistry-climate models as they 1) allow the observed burdens of intermediate lifetime (t>1 year) species to be imposed, bypassing the need to
include emissions, and 2) remove the need for a long spin up for longer lived species burdens to reach a steady state. By using a LBC any chemical feedbacks that would affect the burden of these species are over-written. However, the effect of feedbacks on the steady state burden can be computed through quantification of the appropriate chemical feedback factors (see e.g., Heimann et al. (2020) for a discussion on how we have done this for methane).

### 2.2 Box model

Atmospheric box model simulations were performed using a coupled $H_2$-$CH_4$-CO-OH chemical scheme to link lower boundary condition values of $H_2$ with fugitive $H_2$ emission rates and obtain $CH_4$ lower boundary conditions for use in the UKESM1 experiments. In addition, the box model was used for extensive testing of the framework for GWP calculations.

Our box modelling approach is described in more detail in Warwick et al. (2022) and is similar to that described in Prather
(1994) and used in other studies (Heimann et al., 2020; Nisbet et al., 2020). The model was initialised with realistic values for total methane emissions (585 Tg($CH_4$) per year), CO (1300 Tg(CO) per year) and $H_2$ (80 Tg($H_2$) per year) (e.g. Saunois et al., 2020; Pieterse et al., 2013; Zheng et al., 2019), and was found to give methane and $H_2$ levels in broad agreement with present day levels – 1865 parts per billion (ppb) $CH_4$, 552 ppb $H_2$ and 101 ppb CO, as well as a $CH_4$ lifetime of 9.9 years, which is within the range of observationally-derived values.




### 2.3 Hydrogen economy scenarios

#### 2.3.1 Changes in the abundance of atmospheric $H_2$ in a hydrogen economy

The increase in abundance of atmospheric $H_2$ in a future hydrogen economy is not well constrained due to uncertainties in fugitive emissions, which will depend on the ultimate size of the hydrogen economy and $H_2$ leakage rates, as well as uncertainties in the amount of $H_2$ undergoing uptake by soils. To guide our model scenarios, we estimate how emissions of $H_2$ and other species may change in response to a switch from fossil fuel to hydrogen technologies in an illustrative hydrogen economy scenario. In this scenario, approximately 23% of global energy consumption (about 133 EJ) is supplied by hydrogen (BP Energy Outlook, 2020). We here assume that 100% of the final energy consumption of fossil fuels in the buildings sector switches to hydrogen, along with 50% of the final energy consumption of fossil fuels in the transport sector and 10% of the final energy consumption of fossil fuels in the power generation sector. The lower percentages for transport and power generation reflect a potentially smaller role for hydrogen in these energy sectors due to other low carbon alternatives such as electric vehicles and wind and solar power together with alternative storage options such as pumped hydro, batteries and compressed air (e.g. Staffell et al., 2019). New hydrogen technologies are assumed to have the same energy efficiency as the fossil fuel technology they are replacing (Staffell et al., 2019), except for in the transport sector where we assume diesel and petrol vehicles with an average tank-to-wheel energy efficiency of 30% will be replaced by vehicles using hydrogen fuel cells with an average efficiency of 50%. Using a net calorific value of 1 kg $H_2$ = 33.3 kWh, approximately 860 Tg $H_2$ yr$^{-1}$ would be required to provide the energy consumption outlined above.

Data on $H_2$ leakage rates is limited and $H_2$ leakage rates from the infrastructure associated with a hydrogen economy are uncertain (e.g. E4tech, 2019). However, $H_2$ leakage rates will likely be higher than for natural gas owing to the small molecule size of hydrogen, all else being equal. A recent study looking at the US natural gas supply chain indicated natural gas leaks of around 2.3% of gross gas production, which is ~60% higher than the US EPA inventory estimate (Alvarez et al., 2018). $H_2$ leakage rates of 1 and 10% (which represent the range of values used in previous studies e.g. Shultz et al. 2003, Warwick et al. 2004), would lead to additional $H_2$ emissions of 9 and 96 Tg $H_2$ yr$^{-1}$ respectively based on a hydrogen economy supplying 23% of total present day energy consumption.

We bypass some of the uncertainty in $H_2$ leakage rates, the ultimate size of the hydrogen economy and soil uptake rates, by performing a range of hydrogen economy scenarios with fixed $H_2$ lower boundary mixing ratios. We consider scenarios with $H_2$ lower boundary conditions ranging from 500 to 2000 ppb (i.e. increases relative to present day mixing ratios of 0 to 1500 ppb), which we believe span much of these uncertainties (see Table 1). For example, based on the size of a hydrogen economy outlined above, and assuming the magnitude of the soil sink increases in line with the increase in $H_2$ mixing ratios (i.e. a constant deposition velocity), box model simulations indicate $H_2$ lower boundary conditions of 750, 1000 and 1500 ppb represent $H_2$ leakage rates of about 3, 7 and 13% respectively (see Section 2.2 for further details on the box modelling).



However, these $H_2$ lower boundary conditions would also be consistent with other fractions of total energy supplied by $H_2$,
alternative soil sink responses and $H_2$ leakage rates. Use of this range of $H_2$ lower boundary conditions provides a clear signal
in the atmospheric response to increased $H_2$ mixing ratios relative to interannual variability, and allows us to explore the
linearity of the atmospheric response to increasing $H_2$. We regard our 2000 ppb $H_2$ scenario as an extreme end member designed
to assess the linearity of the atmospheric response, rather than a projection of potential future atmospheric $H_2$ levels in a
hydrogen economy.

## 2.3.2 Changes in $CH_4$, CO, $NO_x$ and VOC emissions in a hydrogen economy

To determine the associated changes in $CH_4$, CO, $NO_x$ and VOC emissions, further assumptions are required about the new
technology employed, in addition to the percentage of different energy sectors switching to $H_2$. In the buildings sector, we
assume that $H_2$ will be combusted and $NO_x$ emissions limited such that they remain unaltered (despite the potential for higher
$NO_x$ emissions from the increased flame temperature), whilst emissions of $CH_4$, CO and VOCs are eliminated. In the transport
sector, we assume a 50% reduction in emissions of $CH_4$, CO, $NO_x$ and VOCs based on 50% of oil-based transport being
replaced by hydrogen fuel cells. In the power sector, we assume that stored hydrogen will be used to generate 30% of the
electricity currently generated using natural gas via combustion and $NO_x$ will remain unaltered, whilst emissions of $CH_4$, CO
and VOCs are scaled appropriately. $CH_4$ emissions associated with energy production are also scaled according to the
decreased use of $CH_4$ where it has been replaced by $H_2$. The assumptions above lead to reductions in $CH_4$, CO and $NO_2$
emissions of 43, 259 and 19 Tg yr$^{-1}$ respectively (see Warwick et al. (2022) for further details).

Emission reductions for CO, $NO_x$ and VOCs are determined by applying a uniform global scaling factor, based on the assumed
changes in fossil fuel usage, to emissions from the Gidden et al. (2019) energy sectors where hydrogen is assumed to play a
role. The emission reductions for $CH_4$ are converted to a change in the $CH_4$ lower boundary condition for the UKESM1
simulations using the box model (see Section 2.2). To separate the atmospheric impacts of increasing hydrogen mixing ratios,
the feedback on the $CH_4$ abundance and changes in the emissions of other species emitted by the consumption of fossil fuels,
four different sets of hydrogen economy model simulations are performed as described in Table 1.

### 2.3.3 Model simulations

For the BASE simulation, all boundary conditions are taken from the recently developed CMIP6 dataset and we assume an $H_2$
LBC of 500 ppb. Climatological boundary conditions for sea surface temperatures, sea ice extent and anthropogenic and natural
emissions are averaged over the years 2000-2014. In the first set of hydrogen economy scenarios, we consider changes to the
atmospheric $H_2$ abundance only. In these simulations, the methane abundance is fixed at 1835 ppb (2014 year level), and does
not respond to changes in the methane lifetime. In the second set of hydrogen economy scenarios, we include the methane
response to changes in atmospheric $H_2$ via the impact of $H_2$ on OH and the methane lifetime. The methane LBCs in these
simulations are determined via a series of box model simulations where $H_2$ is varied (see Section 2.2). In the third set of





hydrogen economy scenarios, we consider reductions in emissions of the ozone precursors CO, NO$_x$ and VOCs from reduced fossil fuel use, whilst the methane abundance is fixed at 1835 ppb. The fourth set of hydrogen economy scenarios includes decreases in emissions of CH$_4$, CO, NO$_x$ and VOCs from reduced fossil fuel use, in addition to the feedback of changes in the methane lifetime on the methane abundance. Each simulation was run for 40 years using annually repeating conditions, with

the final 25 years of each simulation used for analysis and the initial 15 years treated as a spin-up period.

| Hydrogen economy scenario | Experiment | H$_2$ LBC (ppb) | CH$_4$ LBC (ppb) | O$_3$ precursor emissions |
|---|---|---|---|---|
| Present day conditions | BASE | 500 | 1835 | 2000-2014 climatology |
| Present day CH$_4$ and present day O$_3$ precursor emissions | 750H2 | 750 | 1835 | 2000-2014 climatology |
|  | 1000H2 | 1000 |  |  |
|  | 1500H2 | 1500 |  |  |
|  | 2000H2 | 2000 |  |  |
| CH$_4$ feedback and present day O$_3$ precursor emissions | 1500H2_CH4f | 1500 | 2058 | 2000-2014 climatology |
|  | 2000H2_CH4f | 2000 | 2171 |  |
| Present day CH$_4$ and reduced O$_3$ precursor emissions | 500H2_O3pre | 500 | 1835 | CO, NO$_x$ and VOC emissions reduced |
|  | 1500H2_O3pre | 1500 |  |  |
|  | 2000H2_O3pre | 2000 |  |  |
| CH$_4$ feedback and reduced O$_3$ precursor emissions | 500H2_CH4f_O3pre | 500 | 1652 | CH$_4$, CO, NO$_x$ and VOC emissions reduced |
|  | 1000H2_CH4f_O3pre | 1000 | 1756 |  |
|  | 2000H2_CH4f_O3pre | 2000 | 1961 |  |

Table 1. List of UKESM1 hydrogen economy simulations employing different H$_2$ lower boundary conditions (LBCs). The
CH$_4$ LBC used in the simulations not including a CH$_4$ feedback is 1835 ppb (2014 year level). In simulations including a CH$_4$ feedback (CH4f), the CH$_4$ LBC is determined using box modelling. Reductions in O3 precursor emissions (O3pre) are described in Section 2.3.2.



## 3 Results

### 3.1 Atmospheric composition impacts following changes in hydrogen and methane

To determine the indirect radiative impact of hydrogen, we need to understand how $CH_4$, $O_3$ and $H_2O$ will change, for unit emission of $H_2$. Those changes will depend not just on how $H_2$ changes but also on the changes of the co-emitted species. The scenario space is particularly complex and our scenarios should not be regarded as predictions. Instead, here we try to indicate generally how the composition of radiatively active species might change and which changes are linear in the change in $H_2$, looking first at the changes driven by $H_2$ and $CH_4$.


Figure 1 summarises the key composition changes seen for a range of different hydrogen lower boundary mixing ratios when emissions of other species from the fossil fuel industry are held constant (blue and red circles) and when they are reduced (orange and green circles). Decreases in OH, the main atmospheric oxidant, are modelled throughout the troposphere with larger decreases in scenarios with higher $H_2$ abundances (Figure 1a), consistent with R1 being the main atmospheric sink for

$H_2$. The change in OH is linear in $H_2$ both when the surface $CH_4$ mixing ratio is held constant (blue circles) and when it is increased to account for the feedback of changes in OH on the methane abundance (red circles). When including this methane feedback, tropospheric mean OH decreases by about $0.90 \times 10^5$ molecules $cm^{-3}$ (or about 10%) for every 1000 ppb (1 ppm) increase in $H_2$ (Figure 1a, red circles). These modelled changes in OH go on to cause a cascade of further composition changes by altering the methane lifetime, production and destruction rates of tropospheric ozone, as well as aerosol nucleation rates

and cloud condensation nuclei (CCN) concentrations.

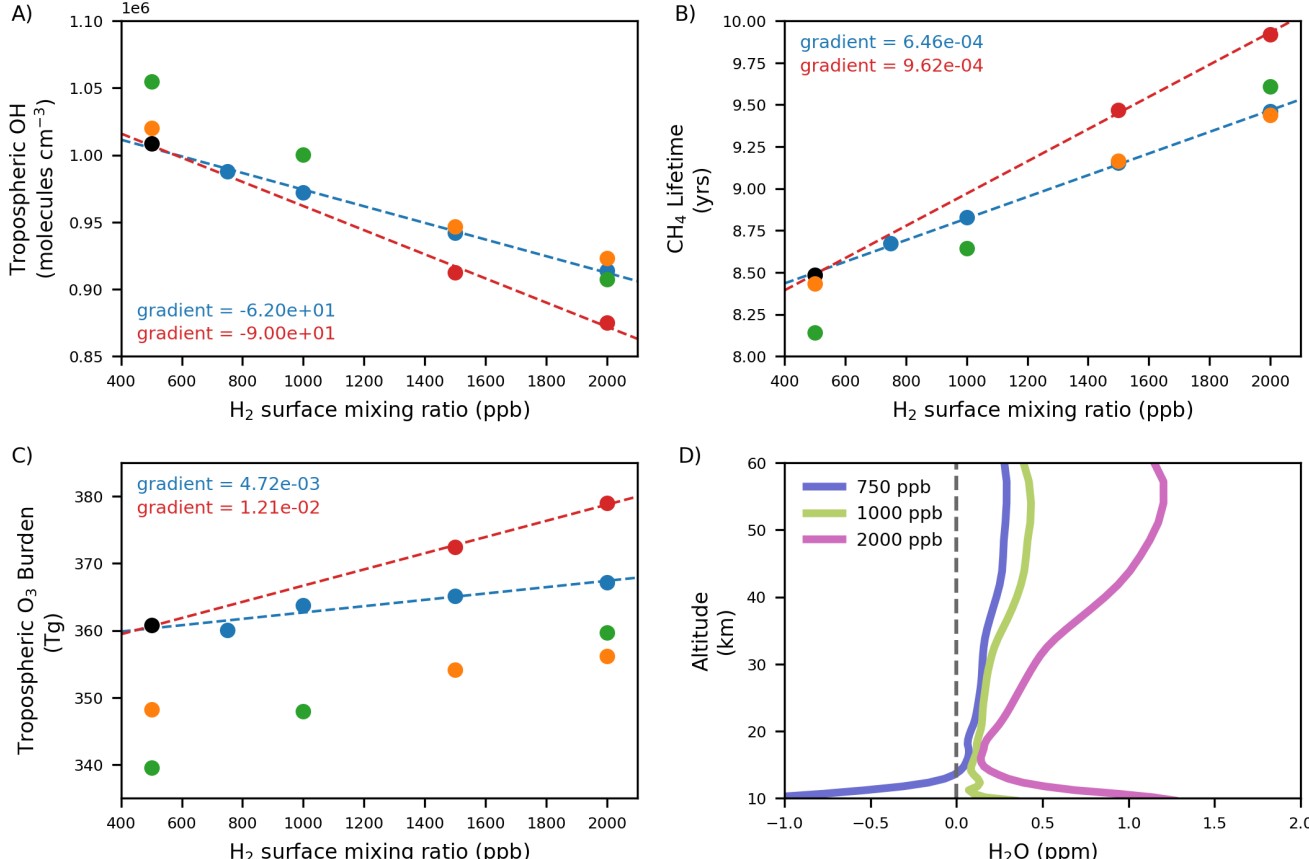

Figure 1. The response of (a) mass weighted tropospheric mean OH, (b) CH₄ lifetime, (c) tropospheric O₃ and (d) stratospheric water vapour to increasing H₂ mixing ratios in UKESM. In Figures 1a-c, the black circle represents the BASE scenario, blue circles represent scenarios where the CH₄ LBC remains fixed at 2014 levels, red circles are scenarios where the CH₄ LBC is adjusted to account for the change in CH₄ lifetime, orange circles are scenarios including changes in emissions of ozone precursors, and green circles include both changes to emissions of CH₄ and ozone precursors, as well as adjusted CH₄ LBCs to account for the response of the CH₄ abundance to changes in the CH₄ lifetime. The blue dashed line represents the fit through experiments where the CH₄ LBC remains fixed at 2014 levels, and the red dashed line is the fit through experiments which include the response of CH₄ to changing H₂ (and OH). Figure 1d shows the change in the vertical profile of H₂O relative to the BASE scenario where the H₂ LBC is increased to 750, 1000 and 2000 ppb (but CH₄ remains fixed at 2014 levels).

Figure 1b shows the CH₄ lifetime as a function of atmospheric H₂ and tropospheric OH. In the BASE simulation, the modelled CH₄ lifetime is 8.5 years, which is in good agreement with other studies (e.g. Stevenson et al. 2013, Prather et al. 2012) and with the lifetime simulated with our box model. As atmospheric H₂ increases but all other factors remain constant, there is a linear increase in the CH₄ lifetime. The methane lifetime increases by 0.64 years for every 1000 ppb (1 ppm) increase in H₂





when the methane abundance is held constant (blue circles), or 0.96 years for every 1000 ppb increase in $H_2$ when the methane abundance is increased in response to the modelled changes in its lifetime (red circles).


The oxidation of $H_2$ in the troposphere can also affect the levels of ozone ($O_3$) as the $HO_2$ produced through R1 can catalyse the interconversion of $NO_x$ which drives tropospheric ozone production (e.g., as reviewed in Archibald et al., 2020a):

$$NO + HO_2 \rightarrow NO_2 + OH \qquad\qquad (R3)$$


followed by photolysis of $NO_2$ to form ozone

$$NO_2 + h\nu \rightarrow NO + O \ (\ +O_2 = O_3) \qquad\qquad (R4)$$

In addition, changes in OH and $HO_2$ produced via R1 influence the destruction of tropospheric ozone by altering the flux through the reaction of $O_3$ with both of these species. Overall, the tropospheric ozone burden increases approximately linearly with atmospheric $H_2$ increase, driven primarily through increases in the reaction $HO_2 + NO$ (R3), which increases by ~7% when atmospheric $H_2$ increases to 2000 ppb (Figure 1c). Changes in the ozone budget are documented in Table S1.

Increased $H_2$ mixing ratios at the surface can also influence stratospheric composition. Increases in $H_2$ and $CH_4$ abundance in the troposphere result in an increased flux of these species to the stratosphere, where their oxidation leads to the production of water vapour. Figure 1d shows the increase in the stratospheric water vapour profile relative to the BASE scenario when tropospheric $H_2$ increases from 500 ppb to 750, 1000 and 2000 ppb, but the tropospheric methane abundance remains unchanged. When $H_2$ increases to 2000 ppb, stratospheric $H_2O$ increases by up to 25% (> 1 ppm). If the tropospheric methane 230 abundance is increased in line with modelled changes in the $CH_4$ lifetime, larger increases in stratospheric $H_2O$ are simulated.

## 3.2 Atmospheric composition impacts when hydrogen, methane and ozone precursor emissions all change

The impacts discussed above arise from changes in atmospheric $H_2$ abundance and subsequent changes in methane abundance arising from the feedback of OH on the methane lifetime. Expected reductions in ozone precursor emissions of CO, $NO_x$ (which may also increase locally), VOCs and $CH_4$ following a shift to a global hydrogen economy also have the potential to 235 further impact atmospheric composition. The green and orange circles in Figures 1a-c show the impact of including these other emission changes, with and without the feedback on the methane abundance included respectively. The relationship remains linear for the airmass weighted tropospheric mean OH, methane lifetime and tropospheric ozone burden plotted as a function





of $H_2$ LBC. In all cases, the gradients remain similar to the corresponding scenarios not including the co-emission reductions (note the ozone precursor emission changes are the same across all scenarios).


The linear relationships shown in Figure 1 allow the global methane and ozone changes for a range of different hydrogen economy scenarios to be estimated, giving the response to different realisations of a future hydrogen economy. They can also be employed to derive a GWP for hydrogen (see Section 3.4).

Some detail on geographical changes in tropospheric ozone in a hydrogen economy is provided in Figure 2. As the photochemical production of ozone is driven by oxidation of carbon monoxide (CO), methane ($CH_4$) and volatile organic compounds (VOCs) in the presence of the oxides of nitrogen ($NO_x$), the net regional change in ozone depends not only on how $H_2$ changes, but also on changes in emissions of these species. This sensitivity is demonstrated in Figure 2 which shows tropospheric ozone column for the BASE scenario, as well as the change in tropospheric ozone column relative to BASE for

different hydrogen economy scenarios, where the tropospheric ozone column is defined as the vertically integrated ozone column between the surface and modelled tropopause. In these simulations we use a blended isentropic-dynamical tropopause combining the 380 K and 2 PV surfaces (following Hoerling et al., 1993). When $H_2$ increases, so does tropospheric ozone (Figures 2c to 2f, see also Figure 1c). Similarly, when atmospheric methane increases, the effect is to enhance the tropospheric ozone column increase (compare Figure 2b with Figure 2a). In contrast, reductions in ozone precursor and $CH_4$ emissions

would avoid the significant $O_3$ increases seen in Figure 2b (see Figures 2g, 2h, 2i and 2j). For example, if there is no $H_2$ leakage, reductions in the other emissions (Figure 2g and 2h) lead, as expected, to reductions in tropospheric ozone. When surface $H_2$ reaches 2000ppb, reductions in the other emissions still lead to modest net decreases in ozone (Figures 2i, 2j). For example, in the UKESM1 simulation which assumes a large $H_2$ leakage, with surface mixing ratios of $H_2$ reaching 2000 ppb, and emissions reductions in CO, $NO_x$, non-methane hydrocarbon and $CH_4$ emissions, the global mean tropospheric column ozone response

is found to be small (-0.1 DU) due to the competing effects outlined above (Figure 2j). The tropospheric ozone response to a global shift towards a hydrogen economy is therefore strongly influenced by the amount of hydrogen added to the atmosphere through leaks, and the co-benefit reductions achieved in CO, $NO_x$, non-methane hydrocarbon and $CH_4$ emissions. Tropopause height differences between the simulations explain only a tiny fraction of the global mean difference in tropospheric ozone column seen (up to 0.06 DU), confirming that the majority of the changes seen here are driven by modelled composition

changes.

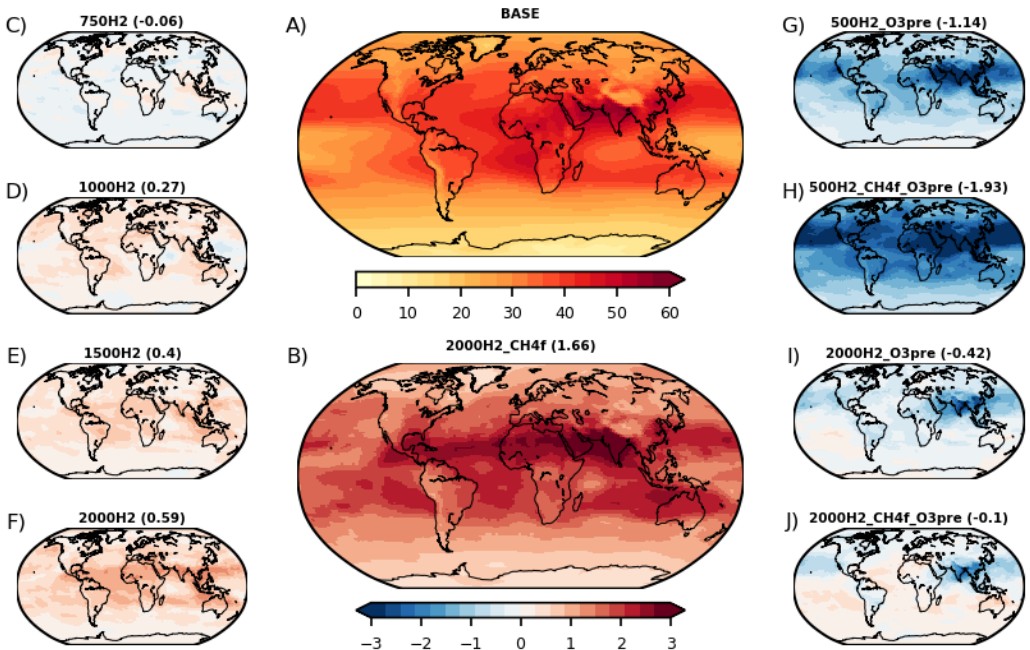

Figure 2. The effect of increases in atmospheric $H_2$ on tropospheric ozone column: (a) tropospheric ozone column for the BASE scenario (DU), (b) the change in tropospheric ozone column relative to BASE when $H_2$ is increased to 2000 ppb, including the response of the methane abundance to changes in the methane lifetime (DU), (c) to (f) changes in tropospheric ozone column relative to BASE when $H_2$ mixing ratios are increased, but all other factors including the methane abundance remain fixed, and (g) to (j) changes in tropospheric ozone column relative to BASE for scenarios including emission reductions in ozone precursors and methane. The numbers above each figure give the change in global mean tropospheric column in DU relative to BASE.

As shown above, the changes in $H_2$ and methane affect OH, which can in turn also influence aerosols in our model simulations. By reducing the flux through the $OH+SO_2$ reaction, there is a shift in the oxidation of $SO_2$ in the gas phase (which contributes to new particle formation) and oxidation in the condensed phase (which grows existing particles). Such a shift leads to a reduction in the aerosol number and increase in aerosol size with impacts on clouds and radiative forcing (see O'Connor et al., 2022 for further details). We simulate a linear decrease in tropospheric air mass weighted OH (Figure 1a), and thus the $OH+SO_2$ reaction flux, for increasing $H_2$ in the atmosphere. This results in a reduction of the gas phase oxidation of $SO_2$ and a reduction in the nucleation of new particles. This shifts the oxidation of $SO_2$ to be favoured by in cloud oxidation, which favours the growth of cloud condensation nuclei. This decrease in OH is coincident with an increase in $HO_2$ (R1 leads to the production of H atoms which in the troposphere near instantaneously form $HO_2$); thus, $H_2$ acts to decrease the $OH$-to-$OH_2$ ratio, similar to CO. The increase in $HO_2$ outweighs the decrease in OH such that the $HO_2$:OH ratio linearly increases with increasing $H_2$ in


the atmosphere (from 145 at 500 ppb $H_2$ to 175 at 2000 ppb $H_2$) and as a result the $H_2O_2$ concentration increases, further enhancing in-cloud oxidation of $SO_2$.

## 290   3.3 Effective radiative forcing

Figure 3 shows the effective radiative forcing (ERF) for the scenarios in Table 1. We determined the ERF using Equation (5) from O'Connor et al., (2021), which can be decomposed into the cloud radiative effect (CRE) and the clear-sky forcing. The time-slice experiments using an atmosphere-only model, i.e. using decoupled sea surface temperature (SST) and sea-ice coverage, permit the determination of the top-of-atmosphere radiative forcing including rapid adjustments to cloud and water

vapour. The ERF was calculated relative to the 2014 base case. The ERF varies from a cooling (negative ERF) to a warming (positive ERF) tendency depending on the scenario. The ERF increases with $H_2$ over the range 500 – 2000 ppb. Table 2 shows that these increases are due to both clear-sky and cloud radiative forcing. The clear-sky forcing increases with increasing $H_2$ presumably due to increased $O_3$, while the CRE can be ascribed to small decreases in cloud albedo. Figure 4 shows that increasing $H_2$ from 500 ppb to 2000 ppb leads to changes in cloud droplet number concentration (CDNC) across the globe.

The decreased CDNC leads, for the same change in water vapour, to larger cloud droplets and lower cloud albedo. This leads to a CRE of 0.05 $Wm^{-2}$. In scenarios where $CH_4$ is increased, the further suppression of OH leads to a stronger CRE. Whilst the radiative forcing from the CRE in our hydrogen economy scenarios suggests this could form a significant component of indirect forcing from $H_2$, we do not include forcing from clouds in our GWP calculations. Such couplings between oxidants, aerosols and clouds are relatively unique in UKESM compared with other CMIP6 models and more studies are required to

constrain the uncertainties involved.




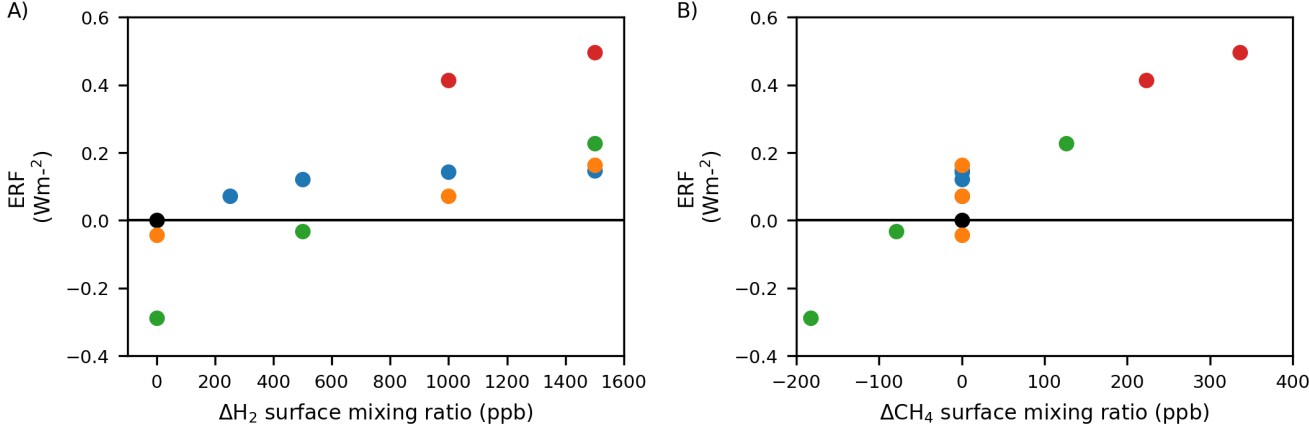

Figure 3. The effect of the simulated changes in atmospheric composition on effective radiative forcing (ERF). Left hand panel shows the ERF as a function of $H_2$ and other experimental conditions; right hand figure shows the ERF as a function of $CH_4$ and other experimental conditions. As for Figures 1a-c, the black circle represents the BASE scenario, blue circles represent

scenarios where the $CH_4$ LBC remains fixed at 2014 levels, red circles are scenarios where the $CH_4$ LBC is adjusted to account for the change in $CH_4$ lifetime, orange circles are scenarios including changes in emissions of ozone precursors, and green circles include both changes to emissions of $CH_4$ and ozone precursors, as well as adjusted $CH_4$ LBCs to account for the response of the $CH_4$ abundance to changes in the $CH_4$ lifetime.

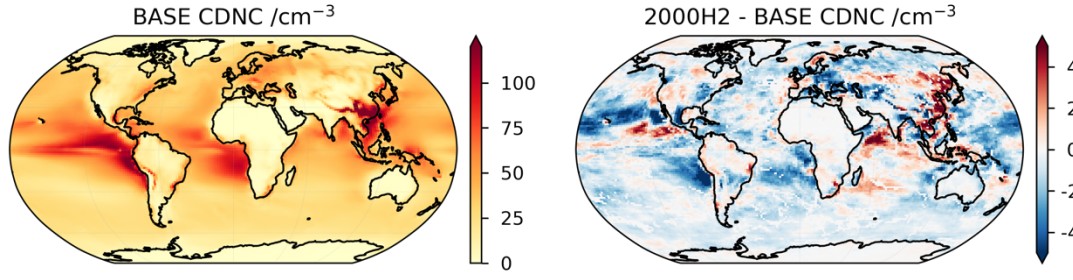


Figure 4. Cloud droplet number concentration (CDNC) in the UKESM1 model at 1 km altitude. The left hand figure shows CDNC in the BASE (500 ppb $H_2$) scenario, the right hand the change in CDNC when $H_2$ is increased to 2000 ppb (2000H2 scenario).





| Hydrogen economy simulation | ERF Clear Sky LW + SW | Cloud Radiative Effect (CRE) LW + SW | ERF all sky, LW+SW |
| --- | --- | --- | --- |
| 750H2 | 0.051 ± 0.041 | 0.021 ± 0.041 | 0.0722± 0.095 |
| 1000H2 | 0.069 ± 0.054 | 0.033 ± 0.054 | 0.102 ± 0.097 |
| 2000H2 | 0.096 ± 0.056 | 0.051 ± 0.056 | 0.148 ± 0.076 |
| 500H2_O3pre | -0.0021 ± 0.053 | -0.051 ± 0.053 | -0.053 ± 0.077 |
| 2000H2_O3pre | 0.110 ± 0.056 | 0.054 ± 0.044 | 0.164 ± 0.087 |
| 2000H2_CH4f_O3pre | 0.186 ± 0.108 | 0.0404 ± 0.108 | 0.223 ± 0.202 |
| 500H2_CH4f_O3pre | -0.160 ± 0.140 | -0.123 ± 0.104 | -0.289 ± 0.196 |
| 2000H2_CH4f | 0.342 ± 0.099 | 0.155 ± 0.104 | 0.497 ± 0.179 |


Table 2. A summary of the effective radiative forcing (ERF) changes in the hydrogen economy simulations.

### 3.4 Global Warming Potential

A GWP quantifying the indirect radiative forcings associated with $H_2$ has so far only previously been determined in a very
limited number of studies including Derwent et al., (2020) (an update on Derwent et al., 2006), Field and Derwent, (2021) and
Hauglustaine et al. (2022). The first two studies considered only the influence of $H_2$ on tropospheric composition, whereas
Hauglustaine et al. (2022) included both the tropospheric and stratospheric response. In addition to these studies, Ocko and
Hamburg (2022) extended the work presented here and in Warwick et al. (2022) to calculate hydrogen's GWP for time horizons
between 0 and 100 years, considering both a pulse of $H_2$ emissions and a constant emission rate.


We determine a GWP for $H_2$ based on composition changes in both the troposphere and stratosphere in our UKESM1
simulations, using radiative forcing scaling factors from IPCC (2014) and IPCC (2021) and the equations E1 to E4 presented
in Section 3.4.1 (see also Warwick et al., (2022)). Our new estimate of the $H_2$ GWP considers changes in methane and ozone
in the troposphere, as well as changes in stratospheric water vapour and stratospheric ozone. Note that we have not used the
model-derived ERFs from Section 3.3 in the GWP calculations below. The model-derived ERFs (for $H_2$ leakage only, including
the methane response) are larger than the radiative forcing due to hydrogen determined from modelled changes in chemical



composition using radiative forcing scaling factors from IPCC (2021) (see Section 3.4.2). Much of this difference arises due to the large contribution from clouds to the model-based ERFs (see Table 2) that is not accounted for in the radiative forcing scaling factors. For our GWP calculations, we determine the radiative forcing using the modelled changes in composition along with the radiative forcing scaling factors rather than the ERFs. Our reason for this is two-fold. Firstly, it ensures the large uncertainty associated with the aerosol-cloud component of the ERFs (IPCC, 2021) is not propagated into the GWP calculations. Secondly, significant further work would be required to attribute the calculated total ERFs to individual species, so that the appropriate lifetimes could be applied to the different components of the forcing that contribute to the GWP. However, we note that if the model ERFs were to be used in the GWP calculations, the GWPs presented in the next section would be larger.

### 3.4.1 Methodology

For our GWP calculation, we derive a more universal version of the approach presented in Fuglestvedt et al. (2010) for calculating GWPs for species whose emissions result in indirect radiative forcings, which was also implicitly adopted for Global Temperature Change Potentials in the IPCC Sixth Assessment Report (IPCC, 2021). The Fuglestvedt et al. (2010) method assumes that the time evolution of the radiative forcing depends only on the lifetime of the species causing the forcing and not the lifetime of the species being emitted. In their method, the time evolution of the radiative forcing during a one-year constant emission is described by an approach to steady-state with a time constant according to the lifetime of the species causing the forcing. The peak forcing is assumed to occur directly at the end of the one-year emission, and subsequently to decay with the same lifetime as in the growth phase. Although this assumption may be a good approximation for short-lived emission species such as $NO_x$, it may not be for species with longer lifetimes such as $H_2$. Our extension of their method (see Warwick et al., 2022) accounts for both the lifetime of the emitted species and the lifetime of the species causing the forcing; it also allows the time period of the constant emission to be varied.

In our method for deriving indirect GWPs from steady-state simulations, we include the indirect forcing due to perturbations in radiatively active species arising (a) during the constant emission (AGWP1, E1), (b) from the decay of the perturbation subsequent to the end of the constant emission (AGWP2, E2) and (c) perturbations arising from the emission species remaining in the atmosphere after the end of the constant emission (AGWP, E3). Derivations for E1 to E3 are presented in Appendix 1.

$$AGWP1 = \frac{R\, a_M \alpha_M \alpha_H C}{t_p} \left( tp - \alpha_M \left( 1 - exp\left(\frac{-t_p}{\alpha_M}\right) \right) - \left(\frac{\alpha_H}{\alpha_H - \alpha_M}\right) \left( \alpha_H \left( 1 - exp\left(\frac{-t_p}{\alpha_H}\right) \right) - \alpha_M \left( 1 - exp\left(\frac{-t_p}{\alpha_M}\right) \right) \right) \right)$$

(E1)


$$AGWP2 = \frac{\left(R\, a_M \alpha_H^2\, \alpha_M\, C\left(1 - \exp\left(\frac{-t_p}{\alpha_H}\right)\right)\right)}{t_p\,(\alpha_H - \alpha_M)} \left(\alpha_H \left(1 - \exp\left(-\frac{(Hz - t_p)}{\alpha_H}\right)\right) - \alpha_M \left(1 - \exp\left(-\frac{(Hz - t_p)}{\alpha_M}\right)\right)\right)$$

(E2)

$$AGWP3 = \frac{R\, a_M \alpha_M^2 \alpha_H\, C}{t_p}\left(\left(1 - \exp\left(-\frac{t_p}{\alpha_M}\right)\right) - \left(\frac{\alpha_H}{\alpha_H - \alpha_M}\right)\left(\exp\left(-\frac{t_p}{\alpha_H}\right) - \exp\left(-\frac{t_p}{\alpha_M}\right)\right)\right)\left(1 - \exp\left(-\frac{(Hz - t_p)}{\alpha_M}\right)\right)$$

(E3)

Where:

$AGWP$ = absolute global warming potential (W m$^{-2}$ kg$^{-1}$ yr)

$M$ = species resulting in the indirect radiative forcing: CH$_4$, O$_3$ and H$_2$O (ppb or DU for tropospheric O$_3$)

$R_M$ = radiative forcing efficiency for M (W m$^{-2}$ ppb$^{-1}$ or W m$^{-2}$ DU$^{-1}$ for tropospheric O$_3$)

$a_M$ = production rate of M [ppb yr$^{-1}$] per ppb H$_2$ change at steady-state (yr$^{-1}$)

$\alpha_M$ = Atmospheric lifetime of M (yr)

$\alpha_H$ = Atmospheric H$_2$ lifetime (combined chemical and deposition lifetime) (yr)

$Hz$ = the time horizon considered (yr)

$C$ = conversion factor for converting H$_2$ mixing ratio (ppb) into H$_2$ mass (kg)

$t_p$ = length of step emission (yr)

The associated indirect GWPs for each radiative species perturbed by hydrogen are then given by:

$$GWP_{H2} = \frac{AGWP_{H2}}{AGWP_{CO2}}$$

(E4)

A comparison of the time evolution of the methane perturbation resulting from a change in H$_2$ mixing ratio based on the standard Fuglestvedt et al. (2010) equations and our updated equations is shown in Figure 5. Accounting for the atmospheric lifetime of hydrogen in the updated equations results in a slower predicted rate of increase in methane during the one-year constant emission relative to the standard Fuglestvedt et al. (2010) equations, a delayed methane peak (peaking at 5 years rather than 1 year, using the parameters specified in the caption) and a slower rate of methane decrease subsequent to that peak; as methane is still being impacted by the remaining H$_2$. This difference occurs as the equations from Fuglestvedt et al. (2010) assume that the emitted species instantaneously reaches a new steady-state atmospheric concentration at the start of the one-year constant emission. In addition, their assumption that the decay following the peak forcing is controlled only by the





lifetime of the radiatively active species neglects any subsequent perturbations to atmospheric chemistry as a result of the emitted species still present in the atmosphere after the end of the one-year constant emission.


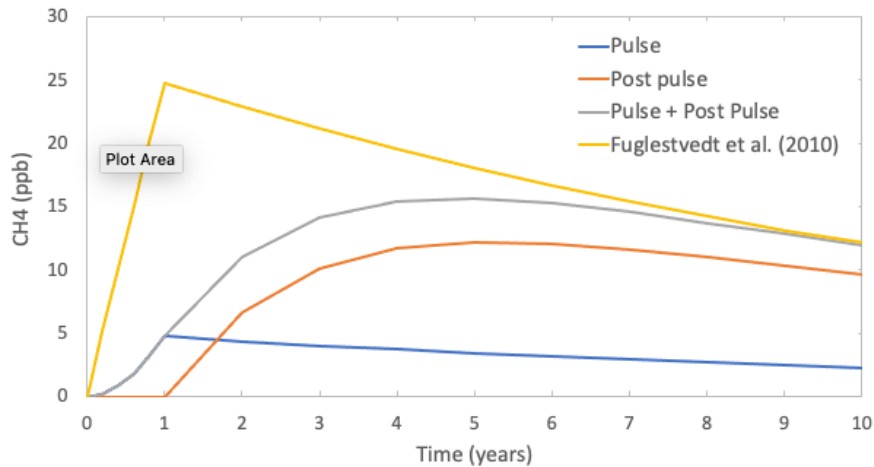

Figure 5. The time evolution of excess methane during and after a one year step hydrogen emission (based on a steady-state H$_2$ excess of +1500 ppb, an H$_2$ lifetime of 1.9 years and a methane perturbation lifetime of 12.6 years) as predicted by the Fuglestvedt et al. (2010) equations (yellow) and our updated equations. For our updated equations, the excess methane is split into contributions from methane generated during the step emission (blue), methane generated from excess hydrogen remaining in the atmosphere after the one year step emission (orange), with the total methane excess shown in grey.

### 3.4.2 Calculation of a hydrogen GWP

In our calculations, AGWP$_{CO2}$(100) and AGWP$_{CO2}$(20) are taken to be 9.17x10$^{-14}$ ($\pm$ 26%) and 2.49x10$^{-14}$ ($\pm$ 18%) W m$^{-2}$ kg$^{-1}$ yr (IPCC, 2014) respectively. AGWP$_{H2}$ is taken as the sum of AGWP1-3. The values for AGWP$_{CO2}$ (100) and AGWP$_{CO2}$(20) change slightly to 8.95x10$^{-14}$($\pm$ 26%) and 2.43x10$^{-14}$($\pm$ 18%) W m$^{-2}$ kg$^{-1}$ yr in IPCC (2021): the impact of this on our H$_2$ GWP calculation, along with other IPCC (2021) updates, is shown in Table 3. The length of the step emission, $t_p$, is 1 year, $Hz$ is taken to be 20 or 100 years and $a_H$ is 1.9 years (with an uncertainty range from 1.4 to 2.2 years). The conversion factor, $C$, for converting H$_2$ mixing ratio into H$_2$ mass based on UKESM1 data is 3.52x10$^{-9}$ ppb kg$^{-1}$. Values for $R_M$, $a_M$, and $\alpha_M$ are dependent on the species causing the indirect forcing (methane, ozone and stratospheric water vapour).

For methane, $R_{CH4}$ is taken to be 0.000363 ($\pm$10%) W m$^{-2}$ (ppb CH$_4$)$^{-1}$ (IPCC, AR5) or 0.000389 W m$^{-2}$ (ppb CH$_4$)$^{-1}$ (IPCC, AR6). The methane perturbation lifetime, $\alpha_{CH4}$, in UKESM1 is derived to be 12.6 years in this study, which falls within the range given by IPCC, AR5 (12.4 $\pm$ 1.4 years). The value for $\alpha_{CH4}$, an additional methane production rate term per unit increase





in $H_2$, represents the positive $CH_4$ tendency resulting from an increase in $H_2$ and corresponding decrease in OH and is defined as:

$$a = CH_4 \, k_{CH4+OH} \, \frac{dOH}{dH_2}$$

where $CH_4$ is the methane lower boundary condition (1835 ppb in UKCA), $k_{CH4+OH}$ is the global mean airmass-weighted rate

constant for the reaction of $CH_4$ with OH and $dOH/dH_2$ is the rate of change of OH with respect to $H_2$. Values for these parameters derived from UKESM1 simulations are as follows: $k(CH_4 + OH) = 2.8 \times 10^{-15}$ cm$^3$ molecule$^{-1}$ s$^{-1}$, $dOH/dH_2 = 3.68 \times 10^{-9}$ (dimensionless). $k(CH_4 + OH)$ is determined from the gradient of methane lifetime against 1/(global airmass-weighted mean OH), and $dOH/dH_2$ from the gradient of the mass-weighted tropospheric mean OH concentration against $H_2$ surface mixing ratio for the UKESM1 simulations including the feedback on the methane lifetime via an adjusted $CH_4$ lower

boundary condition determined by the box model (see Figure 1a). It is also possible to derive $dOH/dH_2$ using the gradient of the mass-weighted tropospheric mean OH concentration against $H_2$ surface mixing ratio from the UKESM1 simulations where only the $H_2$ LBC is varied (whilst the $CH_4$ LBC is held constant) in combination with the equation below from Stevenson et al. (2013), and the methane feedback factor of 1.49 derived for UKESM1,

$$CH_{4 \, new} = CH_{4 \, base} \left( \frac{\tau_{new}}{\tau_{base}} \right)^f$$

where $f$ is the methane feedback factor, $CH_{4 \, base}$ and $\tau_{base}$ are the lower boundary condition and chemical lifetime for methane in the base scenario, and $CH_{4 \, new}$ and $\tau_{new}$ are the lower boundary condition and chemical lifetime for methane in the perturbed $H_2$ scenarios. The difference between the $dOH/dH_2$ values derived using the two methods above leads to a 2%

difference in the derived hydrogen GWP, which is negligible in comparison to other uncertainties (see Table 3), and our GWP calculation therefore remains independent of box model results.

The indirect forcing from methane-induced changes to tropospheric ozone and stratospheric water vapour are determined by multiplying the calculated direct methane forcing ($R_{CH4}$) by scaling factors of $0.5 \pm 55\%$ and $0.15 \pm 70\%$ respectively (IPCC,

2014). Equivalent scaling factors from IPCC (2021) are $0.37 \pm 50\%$ and $0.106 \pm 100\%$.

In addition to the methane-induced indirect radiative forcing from ozone and water vapour outlined above, we also consider non-methane induced forcing from $H_2$ as a source of tropospheric ozone and stratospheric water vapour. The e-folding lifetime of the water vapour perturbation following a change in $H_2$, $\alpha_{H2O}$, is derived from model spin-up data following a change to

the $H_2$ lower boundary condition and is determined to be 8 years in the lower stratosphere. A value for $R_{H2O}$ is determined via comparison of our results to that of Myhre et al. (2007). The vertical profile of stratospheric water vapour changes relative to



BASE obtained for our 2000H2 scenario (Figure 1d) shows very similar changes at all altitudes with stratospheric water vapour changes resulting from methane changes between 1950 and 2000 in Figure 2b of Myhre et al. (2007). We therefore adopt the radiative forcing determined by Myhre et al. (2007) of 0.05 W m$^{-2}$ with an estimated uncertainty of ±20% for stratospheric

water vapour changes between our BASE and 2000H2 scenario. This water vapour forcing is associated with a change of ~500 ppb in stratospheric water vapour at 30 km altitude, giving a value for $R_{H2O}$ of $1 \times 10^{-4}$ W m$^{-2}$ (ppb H2O)$^{-1}$ when considering changes at 30 km. The choice of 30 km for determining $R_{H2O}$ is arbitrary as we assume that the entire stratospheric H2O profile will scale proportionally with the H2 lower boundary condition (following results from our UKESM1 simulations using H2 LBCs of 750, 1000 and 2000 ppb H2). The production rate of water vapour per ppb change in H2, $a_{H2O}$, is then given by the

change in steady-state water vapour at 30 km (500 ppb) divided by the lifetime of 8 years and the change in H2 mixing ratio of 1500 ppb.

To determine the contribution of changes in tropospheric ozone to the hydrogen GWP, the e-folding lifetime associated with the ozone perturbation, $\alpha_{O3}$, is taken to be 0.07 years, calculated as the tropospheric O3 burden divided by the loss in our

UKESM1 BASE simulation. This is shorter than the lifetime used for the short-lived ozone perturbation in Fuglestvedt et al. (2010) of 0.267 years. However our results are insensitive to the difference between these values and small variations around them. The value for $R_{O3}$ is 0.042 (0.037 to 0.047) W m$^{-2}$ DU$^{-1}$ and is taken directly from IPCC, (2014). No updated value for $R_{O3}$ was presented in IPCC (2021), although a similar value of 0.039 W m$^{-2}$ DU$^{-1}$ can be derived based on the estimated change in tropospheric ozone of 109 ± 25 Tg between 1850 and 2005 and a stratospheric-temperature-adjusted radiative forcing due

to tropospheric ozone of 0.39 ± 0.07 W m$^{-2}$ between 1850 and 2010. The value for $a_{O3}$ is calculated as the change in the steady-state global mean tropospheric ozone column in our 2000H2 simulation relative to base (0.59 DU) divided by the lifetime of 0.07 years and the change in H2 mixing ratio of 1500 ppb.

Changes in H2 mixing ratio can also influence stratospheric ozone. The change in global mean stratospheric ozone column between our BASE and 2000H2_CH4f scenarios was +0.10 DU, relative to a base value of 283.78 DU and interannual

variations of up to several DU. Taking a value for $R$ for stratospheric ozone of 0.0054 W m$^{-2}$ DU$^{-1}$ (Schwarzkopf and Ramaswamy, 1993), and a range of e-folding lifetimes for the stratospheric ozone perturbation of between 1 and 10 years, this results in a hydrogen GWP(100) due to indirect changes in stratospheric ozone of less than 0.03 using equations E1 to E4. This value is equivalent to approximately 0.3% of the total GWP(100) determined for hydrogen and is negligible in comparison to

our estimated uncertainties in the hydrogen GWP due to uncertainties in the radiative forcing scaling factors and the soil sink (see Table 3). We therefore conclude that changes in stratospheric ozone for our range of scenarios do not significantly contribute to the hydrogen GWP.

For a 100 year time horizon we determine a H2 GWP of 12 ± 6 (see Table 3). This value is larger than previously published

studies which do not account for changes in stratospheric composition (Derwent et al., 2020, 5 ± 1, and Derwent and Field



2021, 3.3 ± 1.4), but similar to the value of Hauglustaine et al. (2022) which includes a tropospheric and stratospheric response (12.8 ± 5.2). Approximately two thirds of our GWP arises from the influence of $H_2$ on methane and ozone distributions in the troposphere and one third from the influence of $H_2$ on stratospheric water vapour. Results from our study can also be compared with that of Paulot et al. (2021), who estimated the indirect radiative forcing at steady-state due to an increase in $H_2$. Their

study accounted for changes in methane, tropospheric ozone and stratospheric water vapour and found an indirect radiative forcing arising from $H_2$ of 0.13 mW m$^{-2}$ ppb$(H_2)^{-1}$. We obtain a larger value of 0.18 ± 0.03 mW m$^{-2}$ ppb$(H_2)^{-1}$ (the same value is obtained using both the IPCC (2014) and IPCC (2021) radiative efficiency scaling factors), which may be partly explained by the larger methane feedback factor in UKESM1 (1.49) relative to the chemistry-climate model used in Paulot et al. (2021). The UKESM1 methane feedback factor lies at the high end of the range of feedback factors reported in Thornhill et al. (2021)

for the pre-industrial, but there are indications from the literature that this value varies with climate state and methane burden (Holmes, 2018).

Uncertainties in our calculation are based on uncertainties in the hydrogen lifetime with respect to soil uptake, as well as uncertainties in radiative forcing scaling factors and the AGWP for $CO_2$. We do not account for uncertainties in chemical

lifetimes and other UKESM1-derived quantities which are likely to vary when using different atmospheric models with different chemistry schemes. In addition, we also note that the sensitivity of ERF in UKESM1 to changes in methane is approximately 65% larger than that indicated by the radiative scaling factors used in the empirical relationships applied in the GWP calculation (O'Connor et al., 2022). This suggests a further source of uncertainty than accounted for by the radiative forcing scaling factor uncertainties outlined in IPCC (2104; 2021), associated with the response of both aerosols and clouds,

as well as ozone and water vapour to changing methane levels.







| Radiatively active species | H$_2$ GWP(100) IPCC AR5 | H$_2$ GWP(100) IPCC AR6 | H$_2$ GWP(20) IPCC AR5 | H$_2$, GWP(20) IPCC AR6 |
|---|---|---|---|---|
| Methane | 4.9 | 5.3 | 13.4 | 14.8 |
| Tropospheric ozone (methane and non-methane induced) | 3.6 | 3.2 | 11.2 | 9.9 |
| Total Troposphere | 8.5 | 8.5 | 24.6 | 24.7 |
| Stratospheric water vapour (methane and non-methane induced) | 3.2 | 3.0 | 9.9 | 9.5 |
| Total | 11.7 (6 − 18) | 11.5 (6 − 18) | 34.6 (19 − 51) | 34.2 (19 − 51) |

Table 3. A comparison of the 100 and 20 year time horizon H$_2$ GWPs (GWP(100) and GWP(20) respectively) determined
using radiative efficiencies and the AGWP for CO$_2$ from IPCC AR5 (2014) and from IPCC AR6 (2021). Uncertainties quoted
account for uncertainties in the H$_2$ lifetime with respect to soil uptake, radiative forcing scaling factors and the AGWP$_{CO2}$ only.

## 4 Discussion and conclusions

Leakage of hydrogen associated with a hydrogen economy will result in an indirect global warming, offsetting greenhouse gas
emission reductions made as a result of a switch from fossil fuel to hydrogen. We have presented a methodology for calculating
the indirect GWP of H$_2$, with principal contributions coming from changes in methane, in tropospheric ozone and in
stratospheric water vapour. The GWP depends on the lifetime of H$_2$ in the atmosphere, on the perturbation lifetimes of the
radiatively active species, (e.g., CH$_4$ and O$_3$) and on their effective production rates per unit change in H$_2$. We present values
for these based on calculations with our chemistry-climate model, UKESM1 and a box model. The 100 year GWP for H$_2$,
based on the above, is 12 ± 6, a value somewhat higher than some previous estimates. Whilst our GWP uncertainty accounts
for uncertainties associated with the size of the soil sink and radiative forcing scaling factors, it does not include uncertainties
in model-derived parameters (e.g. chemical and perturbation lifetimes or the chemical production rates per unit change in H$_2$).
In particular, uncertainties associated with the stratospheric response may be significant, as well as the response of aerosols
and clouds to changing methane levels. Further simulations using different earth system models and different chemistry
schemes would be required to fully assess the impact of uncertainties in these parameters on the hydrogen GWP.


Determining a GWP for hydrogen allows a change in hydrogen emissions to be compared to an equivalent change in carbon
dioxide emissions in terms of time integrated radiative forcing. This increase in equivalent carbon dioxide emissions can be
compared with expected reductions in carbon dioxide and methane (as equivalent CO$_2$) emissions to determine the net impact



on radiative forcing. In our illustrative future global hydrogen economy scenario (Section 2.3) we estimate additional $H_2$

emissions of between 9 and 95 Tg $H_2$ yr$^{-1}$, based on a hydrogen economy supplying 23% of present day energy consumption and $H_2$ leakage rates of 1 and 10%. Using a $H_2$ GWP(100) of 12, this is equivalent to the time integrated radiative forcing from carbon dioxide emissions of about 110 and 1140 Tg $CO_2$ yr$^{-1}$ respectively. Based on a reduction in fossil fuel use following the fossil fuel energy sectors that are replaced by hydrogen in Section 2.3.1, a low carbon method of hydrogen generation and the sector emissions described in Hoesly et al. (2018), we would expect a reduction in carbon dioxide emissions of ~26,000

Tg yr$^{-1}$. In addition, expected reductions in methane emissions would lead to a further reduction of ~1200 Tg yr$^{-1}$ equivalent $CO_2$ emissions (assuming a GWP(100) for methane of 28). This gives a total reduction of $CO_2$ equivalent emissions of ~27,200 Tg yr$^{-1}$ in our hydrogen economy scenario. In this case, the increase in equivalent $CO_2$ emissions based on 1% and 10% $H_2$ leakage rate offsets approximately 0.4 and 4% of the total equivalent $CO_2$ emission reductions respectively, with the benefits from equivalent $CO_2$ emission reductions significantly outweighing the disbenefits arising from $H_2$ leakage. However, our

calculations clearly demonstrate the climate importance of controlling $H_2$ leakage within a hydrogen economy.

A switch to a hydrogen economy would also provide the opportunity to reduce emissions of other gases which can themselves directly or indirectly affect both climate and air quality. For example, an immediate impact of increased atmospheric $H_2$ is to reduce the concentration of OH, the major tropospheric oxidant, which would thus tend to increase the lifetime of methane.

Increases in methane would adversely affect climate and also lead to production of tropospheric ozone, impacting both climate and air quality. However, we show that reductions in methane emissions associated with decreased fossil fuel use, along with reductions of CO and $NO_x$ emissions, could lead to a net change in ozone close to (or below) zero with associated climate and air quality benefits, assuming very high $H_2$ leakage rates (e.g. scenario 2000H2_CH4f_O3pre, Figure 2j). In the case of no leakage of $H_2$, emission reductions in methane, CO, and $NO_x$ lead to decreases in tropospheric ozone column globally (scenario

500H2, CH4f_O3pre, Figure 2h). To assess whether the ozone response to changes in emissions associated with a hydrogen economy is influenced by the assumed background climate conditions, (i.e. whether the timing of the implementation of a hydrogen economy could be important), we performed a set of additional model simulations using 2045-2055 climatological boundary conditions taken from the CMIP6-SSP2-4.5 scenario. These scenarios assumed a hydrogen economy of the same absolute size and identical leakage rates as our present day (2014) scenarios. We found no discernible difference in terms of

the tropospheric and stratospheric ozone response to changes in $H_2$ and ozone precursors, with emission reductions from decreased fossil fuel use offering similar air quality benefits in both sets of simulations. To maximise the benefit of a switch to hydrogen, not only should hydrogen leakage be minimised, but the emissions of methane, CO, VOCs and $NO_x$ should also be reduced to the maximum extent possible.



**Author Contributions**

JAP and ATA designed the research and supervised the analysis. JK performed the UKESM model simulations and atmospheric composition analysis and PTG and FMO performed the ERF calculations. KPS, JAP, PTG and NJW contributed to the $H_2$ GWP analysis. KPS derived the AGWP equations, with contributions from JAP. NJW performed the UKESM GWP calculations. All box modelling was performed by PTG. NJW compiled the emission changes for the different scenarios. All authors contributed to the writing of the manuscript.

**Code availability**

Due to intellectual property rights restrictions, we cannot provide either the source code or documentation papers for the UM. The Met Office Unified Model is available for use under licence. A number of research organisations and national meteorological services use the UM in collaboration with the Met Office to undertake basic atmospheric process research, produce forecasts, develop the UM code, and build and evaluate Earth system models. For further information on how to apply for a licence, see http://www.metoffice.gov.uk/research/modelling-systems/unified-model.

**Acknowledgements**

The research presented here was funded by the Department for Business, Energy and Industrial Strategy (BEIS). In addition, the HECTER project supported Alex Archibald, Paul Griffiths, Nicola Warwick (NE/X010236/1) and Keith Shine (NE/X010732/1) in the latter stages of this work. Fiona O'Connor was supported by the Met Office Hadley Centre Climate Programme, funded by BEIS. Model simulations and analysis were performed using Monsoon2, a collaborative High Performance Computing facility funded by the Met Office and the Natural Environment Research Council, JASMIN, the UK collaborative data analysis facility, and the NEXCS High Performance Computing facility funded by the Natural Environment Research Council and delivered by the Met Office. We thank Ken Caldeira and Lei Duan, Carnegie Institution for Science, for their contribution to the derivation of the AGWP equations.

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
