# Peer review of "Atmospheric composition and climate impacts of a future hydrogen economy"

_Atmospheric Chemistry and Physics, 2023_

## Author Comment (AC1)

We would like to thank the anonymous referees and Matteo Bertagni for their useful comments and suggestions which have helped improve the manuscript. We respond below to each individual comment raised by the referees. Referee comments are shown in italic.

The Sand et al. (2023) appeared at a very late stage in the revision processes (on 7 June 2023). As the results of that paper are very relevant to (and in good agreement with) our work, we have added brief mentions to this paper in our revision in addition to changes in response to the referee comments below.

Sand, M., Skeie, R.B., Sandstad, M. *et al.* A multi-model assessment of the Global Warming Potential of hydrogen. *Commun Earth Environ,* 4, 203 (2023). https://doi.org/10.1038/s43247-023-00857-8

**Referee 1**

*This paper is a pleasant-to-read update of the possible role of a molecular H2 economy on the climate through chemical modulation of the greenhouse gases CH4, H2O, and O3. The authors do a great job of discussing complete scenarios in terms of a life-cycle assessment involving the swap out of fossil fuels for H2. It is a single-model result that is valuable but it does not help us choose or consolidate the large range of published results on this topic over the last two decades. It does present an excellent analysis of why Fuglestvedt's short-term approach for H2 did not work, but it does not really contain a "new methodology".*

*The model simulations are all credible and well designed, but I find three major problems with the write-up of their results. These could be readily fixed if the authors choose to do so.*

*(1) Abstract: "We find that increases in hydrogen result in increases in methane, tropospheric ozone and stratospheric water vapour,…" We all knew this 20 years ago (see T. K. Tromp et al., Science 300, 1740 (2003) and M. J. Prather, Science, 302, 581 (2003), also R. G. Derwent et al, Climatic Change, 49, 463 (2001)) and even the old papers estimated the climate impacts. The claim here in the abstract, and as written through the paper, is as though this is your discovery. I am sure that you knew the approximate answers before you began this modeling, so please recognize the literature and your place in it.*

The wording of the abstract has been changed so the impact of hydrogen on methane, tropospheric ozone and stratospheric water vapour is not misconstrued as a new finding. A reference to the Tromp et al. (2003) study was included in the Introduction on line 54, however we have now further included references to the Prather (2003) and Derwent et al. (2001) studies on lines 27 and 44 respectively. We have changed the wording of the abstract from 'we derive a new methodology' to 'we derive a refined methodology'.

*(2) This paper purports to show new results, yet it heavily references many of the approaches and results in a UK government report listed as Warwick et all. 2022. It seems as though much of the current work appeared already in the 2022 report. Is this double publication? I have great sympathy with authors who may have been required to deliver the government report, but then the referencing to it should be much clearer than it is now, and it should note that that 75-page report contains material that is reproduced in the standard scientific literature here. I greatly favor this work coming out in ACP, but make it clear to the readers.*

The online UK government report containing some of the work presented here was not peer-reviewed and this is therefore the first publication of the work in peer-reviewed literature. A

sentence has been added towards the end of the introduction (L67-68) stating that the work presented in this manuscript is based on, and an extension of, the work presented in Warwick et al. (2022) so that this is clear to readers from the start.

*(3) Abstract & Methodology. "We derive a new methodology for determining indirect Global Warming Potentials from steady-state simulations …" The authors apparently do not understand the relationship between steady-state perturbations and integrated pulse-emission transients which is clearly spelled out in Fuglestvedt 2010 ("Generally, so-called steady-state [GWPs] are used. They are calculated assuming constant emissions and steady-state conditions. For compounds that are removed by linear processes, this is equivalent to assuming an emission pulse and integrating over the entire decay of the compound (Prather, 1996, 2002)."). The two approaches give identical results, and the scaling factor is the steady-state lifetime of the perturbing species (H2 in this case). The mistake made by Fuglestvedt was in assuming that a 1-year constant emission of the perturbing gas reached steady state. For NOx and some VOCs, it did. For CO is did not quite and for H2 it certainly did not. That was a good catch and those H2 GWPs need to be revisited. I am not sure that any modern H2 GWPs rely on the shorthand calculations used in Fuglestvedt 2010.*

The wording in the abstract was not precise enough and we replace it with "We derive a refined methodology for determining indirect Global Warming Potentials **using parameters derived** from steady-state UKESM1 simulations …".

The reviewer's comment, regarding the application of Fuglestvedt et al. (2010) methodology to derive H2 GWPs and the need for their values to be revisited, is a strange one as Fuglestvedt et al. state, very clearly, at the end of their Section 7.3 "We do not provide values for $H_2$" – instead they refer to an earlier paper by Derwent et al..

*The equivalence between steady-state and integrated transient is exact, but the steady-state must include all species. To avoid a 100-year calculation to get CH4 in steady-state, most modelers force a steady state in a few years (CO time scale, and maybe O3 in the stratosphere) with H2 and CH4 fixed/perturbed at the lower boundary. We then use the change in budget, along with feedback factors to project the CH4 and H2 in steady state. This could/should have been estimated for the 1-year emission runs in F 2010. Adding transients to the short-term 1-year steady-state is questionable. I am not sure it gives the correct true steady-state answer or the correct integration of the transient following a single, annual pulse. The derivation here is not convincing as the perturbation must be decomposed into all the chemical modes it excites.*

We do not know what the reviewer means by the "1-year emission runs in F 2010". That paper did not perform any emission runs. Rather they used steady-state calculations presented in the literature to derive the parameters that fed into their GWP methodology.

*Methodology: L345. Correct, we agree it is the budget lifetime of the emitted/perturbing species that determines the scaling. However, the perturbation timescale of the secondary species such as CH4 determines the amplitude of that perturbation.*

*L 353: This is incorrect. If you have a steady state then you should NOT be using the subsequent decay. Period. Look up the steady-state equivalence derivation again (i.e., CH4 must also be in steady-state. If you simply have a pulse emission (even over 1 year), then you must follow ALL transients.*

As in the abstract, it is the wording which is loose. We do not assume steady-state in the GWP calculations, but rather we use the steady state UKESM1 calculations to derive the value for 'a', the production rate of CH4 per ppb change in H2, which is then used in our GWP calculations. We confirm that we do follow all transients in the GWP calculations – hence our Equations E1 to E3 are applied separately to methane and non-methane-induced ozone and water vapour (as now stated in an extra sentence on L425 so that this is clear to the reader).

*L 357-375: If you want to just follow the transients, then the time constants you have here are wrong. The 'atmospheric lifetime' for H2 and CH4 are the budget times: Burden/Loss. But, the decay times here are the chemical mode times (need an eigenvalue decomposition or else an estimate of the perturbation time scale from the feedbacks) and are longer than the lifetime. You do use 'perturbation time' in Figure 5, but not in these equations? You know that H2 changes OH, and so it also has a perturbation time (given the soil sink, this is much closer to the standard lifetime).*

We have used the perturbation lifetime of methane in the equations and have updated the manuscript to make this clear. We did not however calculate a perturbation lifetime for H2, as we believed the impact would be small due to the dominance of the soil sink over the OH sink. We have now calculated a perturbation lifetime for H2, which differs from the standard lifetime by approximately 0.04 years. This has a minor impact on the GWP calculations (approx. difference of 0.4 for GWP(100)), which is offset by the impact of adjustments to the methane feedback factor included in response to the comment on L427, resulting in minimal overall changes to our estimated GWPs (see changes to Table 3).

*L 425: the mass-weighted OH mean is a truly terrible metric for anything. It simply does not describe the CH4 or H2 loss. You should be calculating the OH weighted by exp(-1775/T) and the CH4 abundance*

Our aim here was to derive a global value for $k_{CH4+OH}$ that accounted for atmospheric variations in temperature and the methane abundance, required as input into the equation to calculate 'a' on L421. A global value for $k_{CH4+OH}$ was derived using a plot of the methane lifetimes from our different H2 LBC simulations against 1/(global airmass-weighted mean OH). Using this method, the impact of variations in temperature and the methane abundance on methane loss are included in the value derived for $k_{CH4+OH}$, which is taken to be 1/gradient.

*L 427: Your CH4 feedback factor here is very high, I suspect that this is calculated from only the tropospheric OH loss. You need to add in the stratospheric loss and soil sink (which have no feedbacks). This Stevenson derivation is really awkward and not the original one used in the SAR and TAR. Using the dCH4 perturbation, one calculates the change in loss of CH4:*

*sensitivity = d ln(L-CH4) / d ln(CH4) , ie, for a 10% CH4 increase at the lower boundary, the denominator is ln(1.1). This sensitivity is < 1 because OH decreases with CH4 (or H2) increases. The feedback factor is then formally ff = 1/(1 – sens).*

*For CH4 and H2, the L-CH4 = trop-OH loss + soil loss + strat loss.*

The feedback factor of 1.49 is based on OH loss only. When including all sinks, the feedback factor reduces to 1.39, which is consistent with other studies. As the feedback factor is used to determine the methane perturbation lifetime used in the GWP calculations, using a value of 1.39 relative to 1.49 reduces the hydrogen GWP(100) determined by ~0.3. This small reduction in the GWP is offset by the increase in GWP resulting from the slight

lengthening of the hydrogen lifetime when a perturbation lifetime is determined for hydrogen (see reply to comment about L357-375). We have updated GWP values in Table 3 to reflect both changes.

*Minor comments:*

*L39: please go back to the original references here.*

The following additional references have now been included: Derwent et al., (2001), Schultz et al., (2003), Warwick et al. (2004), Prather et al. (2003) (lines 44-45).

*L175: Very good point about the shift in fossil fuel related SLCFs. Nicely done. This Life cycle assessment is needed.*

*L186: As above, we really do not care about the mass-weighted OH change, it does not tell us about CH4 loss or even CO loss.*

We quote the change in mass-weighted OH per 1 ppm increase in H2 as it is required as an input into our GWP calculations (needed to determine 'a' on L467). Our value for dOH/dH2 is then multiplied by our derived global value for $k_{CH4+OH}$ (which accounts for varying atmospheric temperatures etc., see above reply to comment on L425), and our baseline CH4 mixing ratio to determine 'a', the production rate of methane per unit change in H2.

*Figure 1: Plot the blue dots on top of the orange to see the fit?*

Figure 1 has been updated to show the blue dots plotted on top of the orange.

*L204: As above, does the CH4 lifetime include OH or also strat and soil, otherwise 8.5 is VERY short! as it implies a total lifetime of 7.5 years vs 9.*

The methane lifetime derived in this study reflects only loss by reaction with OH. Although this value implies a total chemical methane lifetime shorter than the range quoted in IPCC AR6 (9.7 +/- 1.1 years), it is more comparable to a multi-model present day whole atmosphere chemical lifetime of 8.4 +/- 0.3 years obtained by Stevenson et al. (2020). Stevenson et al. found significantly shorter methane lifetimes for the present day compared to pre-industrial, mainly due to increases in model OH. We have updated the text to reflect the above on L228-231.

Stevenson, D. S., Zhao, A., Naik, V., O'Connor, F. M., et al.: Trends in global tropospheric hydroxyl radical and methane lifetime since 1850 from AerChemMIP, Atmos. Chem. Phys., 20, 12905–12920, https://doi.org/10.5194/acp-20-12905-2020, 2020.

*How does strat O3 change in response to the H2O affect photolysis and trop O3?*

It is difficult to separate the direct impacts of hydrogen changes on tropospheric ozone, and those induced by changes to stratospheric ozone via changes to photolysis and strat-trop exchange given the diagnostics available from these experiments. However, we believe that they are small – in a study exploring the tropospheric ozone response to stratospheric ozone depletion associated with ODSs (Griffiths et al., 2020) using the UKCA chemistry scheme (the same scheme used in this study), STE decreased by ~75 Tg yr$^{-1}$, while net chemical production increased by 50 Tg yr$^{-1}$. The stratospheric ozone changes associated with changing hydrogen explored in this study are orders of magnitude smaller than those of the Griffiths study (the change in global mean stratospheric ozone column between our BASE

and 2000H2_CH4f scenarios was +0.10 DU), which would lead us to expect only small changes to STE and tropospheric photolysis in the simulations used here.

Griffiths, P. T., Keeble, J., Shin, Y. M., Abraham, N. L., Archibald, A. T., & Pyle, J. A. (2020). On the changing role of the stratosphere on the tropospheric ozone budget: 1979–2010. *Geophysical Research Letters*, 47, e2019GL086901. https://doi.org/10.1029/2019GL086901

*L260 why not also Fig 1c?  blue vs red*

We believe the referee is referring to Fig 2c here rather than Fig 1c. The small changes in ozone in response to increasing the hydrogen LBC from 500 to 750 ppb in Fig 2c are not significant. Figure 2 has now been updated to only show differences that are statistically significant at the 95% confidence interval.

*L271-283:  This whole section on calculating the impact of OH changes on aerosol formation is so model-dependent that the results and even the sign of the results are simply not robust.  I have no doubt that the authors tried to simulate aerosols in their model and these are their results.  But, let us face the facts:  the processes that drive these aerosol results occur on space and time scales that are in no way resolved with any global model.  The processing of aerosols and SO2 as in liquid clouds in the boundary layer, or in being transported out of the boundary layer is complex and I doubt that many models have the same mechanisms or agree on how to model this.  If given the same task (increase CH4, which changes OH and HO2 and O3), a MIP were to reproduce these results, I would be shocked.*

*For this model result, there are several problems already.  Where are the other aerosols and how do they respond (nitrates? SOAs?).  One of the competition for SO2 is surface deposition and that is long known to compete with OH oxidation to SO3.  This depends on boundary layer mixing not discussed here.  Further, the SO3 probably attaches to existing aerosols in the polluted boundary layer and does not form new particles.  The analysis here argues about mass-weighted OH changes, but it is really only the boundary layer that matters here.  Once in the free troposphere, all of SO2 will probably react with OH since it is unlikely to be pulled into a cloud.*

*The chemistry discussion here is confusing:  "The increase in HO2 outweighs the decrease in OH" makes no sense.  HO2 >> OH and so the HO2:OH ratio is driven by the decrease in OH, not the increase in HO2. e.g., a ratio of 1000:10 goes to 1001:9*

*I do not doubt the results for the UKESM1 model in O'Connor et al (2022) but I doubt that any other model, given the complexity of competing forces, would give the same or close to the same answer.*

We acknowledge the large uncertainty involved in determining aerosol feedbacks. However, we believe there is still value in presenting our results so this can be tested/evaluated in other models and the results compared. We have rewritten parts of this paragraph to make the chemistry discussion clearer and also added a sentence highlighting the need for more model studies with a comprehensive representation of aerosols to constrain the uncertainty on L320: 'Whilst our simulations demonstrate the potential importance of aerosol feedbacks relating to changes in OH, the inclusion of other aerosols, e.g. nitrates, may influence our results and more studies involving multiple models are required to constrain the uncertainties involved.'

*L284ff: Since the UKESM is being run here as an atmosphere model with SSTs, it generates it own weather and cloud systems. The weather/clouds are chaotic and not simply determined by the SSTs. Thus with aerosol and heating rate perturbations, how many decades did you have to run to get a robust signal for things like CRE? Is this whole section on aerosol-cloud forcing meaningful? It is not convincing to me.*

As the reviewer rightly points out, the weather and clouds are chaotic. To account for this variability, the simulations were run for 40 years, with the latter 30 years to derive effective radiative forcings. This length of simulation follows the recommendations by Forster et al. (2016). The robustness of the ERFs and their components can be determined by the standard errors computed, which are of the order of 0.05 W m$^{-2}$ or less. In some cases, the CRE is weak and within the noise, whereas there are other cases where it is larger and outside of the noise.

Forster, P. M., Richardson, T., Maycock, A. C., Smith, C. J., Samset, B. H., Myhre, G., Andrews, T., Pincus, R., and Schulz, M.: Recommendations for diagnosing effective radiative forcing from climate models for CMIP6, J. Geophys. Res.-Atmos., 121, 12460–12475, https://doi.org/10.1002/2016JD025320, 2016.

*L300ff: It would be nice to have tabulated need individual components of ERF and other chemical changes for others to compare with.*

We have recalculated the ERF values using an updated method from Ghan (2013) and also now split them into the LW and SW components in Table 2. We are unable to spilt the ERF into the methane/ozone/water vapour components without performing a series of further model simulations (this will be investigated in future work as part of a newly funded project).

**Referee 2**

*General Comments*

*This paper reports important results on the climate impacts of hydrogen leakage to the atmosphere, using a set of experiments with the UKESM1 model. The study is well designed and presented, and will be of wide interest to both scientists working on this problem and policymakers considering the pros and cons of using hydrogen as a future fuel.*

*I have a few queries and suggestions for clarification and improvement (details below). Once these are addressed, I am happy to recommend publication.*

*Specific Comments*

*Abstract. A key result not mentioned in the abstract is the large cloud adjustment found in the ERF calculations. Whilst uncertain, this result seems potentially very important, as it increases the climate impacts by about 50%, and I think should be included here. I also think the net climate impact (including $CO_2$ reductions associated with the uptake of a $H_2$ economy) should also be clearly stated in the abstract, even though these are only done simply here, and not a crucial part of the study.*

We chose not to include the large cloud adjustment found in the ERF calculations in the abstract due to the potential model dependency and high uncertainty associated with this result (see also replies to comments by Referee 1). However, we believe it is still useful to include our result in the manuscript to highlight the need for further work in this area.

The following sentence has been added to the abstract concerning the net climate impact - "Based on this GWP, and hydrogen leakage rates of 1 and 10%, we find that hydrogen leakage offsets approximately 0.4 and 4% respectively of total equivalent $CO_2$ emission reductions in our global hydrogen economy scenario."

*Figure 1. NB the figure has capital A, B, C, D, but the caption has lower case (also other figures). Suggestion for Figure 1d: the large deviations of water vapor between scenarios below about 15 km (i.e. partly from the troposphere) are not useful, so why not start the profiles a bit higher (~15 km)? Also, it would be interesting to see the water vapor profiles for the simulations with adjusted methane (and $O_3$ precursors) as well as those shown, since the simulations with adjusted methane are the ones that show the most complete overall impacts of $H_2$ emissions.*

The figure caption has been changed to refer to capitals (A, B, C, D). Figure 1 has been updated to start the water vapour profiles at 15 km , and to include water vapour profiles for simulations with adjusted methane. Simulations with adjusted O3 precursors are not included to so that the plot remains clear and does not become overcrowded.

*L205. Is the methane lifetime of 8.5 years a total lifetime (i.e. with respect to OH, Cl, soils and stratospheric losses; Prather et al., 2012) or just with respect to OH (and if the latter, is it both tropospheric and stratospheric OH)? Stevenson et al. (2013) Table 7 reports values for total lifetime.*

The methane lifetime derived in this study reflects only loss by reaction with tropospheric OH and this has been clarified in the text on L228 (see also reply to Referee 1). It is comparable to a multi-model present day whole atmosphere chemical lifetime of 8.4 +/- 0.3 years obtained by Stevenson et al. (2020), albeit this value includes all chemical sinks.

*L253. Interestingly (and the opposite of what you say), Figure 2c (750 ppm $H_2$) actually shows a small net decrease in ozone (this can also be seen in Fig. 1c and Table S1). Is this a significant result (one would expect an increase, based on the other experiments), or is this just illustrating the level of meteorological noise in the simulations?*

Figure 2 has now been updated to only show differences that are significant at the 95% confidence interval. It can be seen that most of the O3 changes in Fig 2c are now masked out and the O3 decrease is therefore not statistically significant.

*L254. Do you mean "compare Figure 2b with Figure 2f" (rather than 2a)?*

The text has been updated to read "compare Figure 2b with Figure 2f".

*L255. I would argue that Figures 2h and 2j are the relevant figures to compare with Figure 2b, and not 2g and 2i, for the context described, since 2g and 2i have fixed methane. (Similar comment on Figure 2 caption, L275.)*

The text has been updated to refer to only Figures 2h and 2j.

*L286 $OH_2$ should be $HO_2$*

The text has been changed to refer to $HO_2$.

*L300 "change in" should be "amount of"?*

"Change in" has been updated to 'amount of'.

*L305 I think it is worth stressing that although these ERF adjustments are apparently only available from UKESM, they do suggest potentially important climate effects via cloud adjustments, and it is important that other modelling groups look at this. The Cloud Radiative Effect seems to add about an extra 50% to the RFs.*

The wording of the following sentence has been changed (L369) to highlight that the CRE forms a significant component of our total ERFs: 'Whilst the radiative forcing from the CRE in our hydrogen economy scenarios **increases the total ERFs by approximately 50%, thus forming a significant component of indirect forcing from $H_2$**, we do not include forcing from clouds in our GWP calculations. Such couplings between oxidants, aerosols and clouds are relatively unique in UKESM compared with other CMIP6 models and more studies are required to constrain the uncertainties involved.'

*Section 3.3 and Figure 3. Is it important to stress here that the ERFs calculated do not include changes in $CO_2$ emissions (but do include changes in $CH_4$ emissions and all $O_3$ precursors, as well as $H_2$)? Later, you do discuss the overall impacts of the scenarios in the context of $CO_2$ emissions, pointing out the overwhelmingly beneficial outcome in terms of impact on $CO_2$/climate of moving to a hydrogen economy (at least a green hydrogen economy). But this should perhaps be absolutely clear in this section too. (It is all too easy for some policymakers to get the wrong end of the stick about $H_2$ and just see a positive ERF or large GWP value and conclude it is a major problem).*

We have now added the following sentences to the start of Section 3.3: 'Our ERF values are based on the chemical changes in composition resulting from increases in atmospheric $H_2$, in addition to changes in emissions of $CH_4$, CO, $NO_x$ and VOCs in the scenarios where these are included. The calculated ERFs do not include the large climate benefit expected from anticipated reductions in $CO_2$.'

Also the following has been added to the caption for Figure 3: 'Note that none of the calculated ERFs include anticipated reductions in CO2.'

*L334 A Global Warming Potential (GWP) with a time horizon of zero years is indeterminable and is meaningless. This is perhaps more a criticism of Ocko and Hamburg (2022), but if this work wants to refer to it, it should warn readers that it is meaningless. The "GWP" calculated in Ocko and Hamburg with a constant emission rate is not the same metric as a GWP (their Figure 3b incorrectly labels it as a GWP), and that should also be clarified, as it is another important source of confusion in the literature.*

The reference to a GWP with a time horizon of zero years has been removed.

*Section 3.4.1 and Figure 5. The formulae for the GWP calculations is certainly complex and opaque (and hence difficult to follow and check), but I have every confidence that the authors have done this correctly, and is probably the best that can be done given the experimental set-up. Some of my confidence comes from a comparison that I feel is worth adding, of Figure 5 and Derwent et al. (2020) Figure 1, which is an equivalent result from a different model that does not constrain methane concentrations with a lower boundary condition. Figure 5 shows a peak response of methane of about 15 ppb for a $H_2$ pulse of 1500 ppb (i.e. ratio of $CH_4$ response to $H_2$ added of ~1/100), with the peak about 5 years after the pulse. Derwent et al. added a pulse of size 1.67 Tg (~6 ppb) $H_2$ to a model with free-to-respond methane – the response is shown in their Figure 1: a peak methane response of ~0.08 ppb (i.e. ratio of ~1/75) after about 3.5 years. So the response represented by the complex equations is similar to that found in a full model simulation*

*(albeit a model with different set-up and different parameters such as $H_2$ and $CH_4$ lifetime, etc.).*

We acknowledge the formulae for the GWP calculations are complex. In addition to equations E1-E3, we now also include a new equation, E5, which allows a simpler approach to be applied with little impact when considering a time horizon of 100 years (see new equation E5 and paragraph at the end of Section 3.4). This equation is the $t_p$ -> 0 version of E2. Using equation E5 rather than equations E1 to E3 makes a negligible difference (less than 0.1%) to GWP(100) using our parameters, but the difference is around 1.5% for GWP(20).

Although not in the original manuscript, we had compared our results with that of Derwent et al. (2020). The following sentence has now been added to the manuscript: 'A comparison of the time evolution of the methane perturbation as determined by equations E1 to E3 in Figure 5 (grey line) with the methane response to an $H_2$ pulse modelled explicitly in a model where methane is free to respond (Derwent et al., 2020), shows that the response represented by equations E1 to E3 is similar to that found in a full model simulation.'

*L497 (and elsewhere). IPCC recommends referring to individual chapters by their lead authors, rather than to the whole reports.*

IPCC references have been updated to refer to the lead authors of the relevant chapters.

*L498. The methane feedback factor for UKESM1 of 1.49 is suspiciously large, and is actually outside the range of models quoted in Thornhill et al. (2021) (the individual models have values of 1.32, 1.31, 1.43, 1.30, 1.26 and 1.19 – and the first one of those is for UKESM1). This makes me think it may have been incorrectly calculated. It is conceivable that the set-up of UKESM1 used here is sufficiently different to change this factor compared to that used in the analysis of Thornhill et al., but that seems unlikely. How important is this feedback factor for the calculated GWP?*

The feedback factor of 1.49 only accounts for the tropospheric OH sink, when including all methane sinks the feedback factor is 1.39. The feedback factor will vary with both methane emissions and climate state, and the value for UKESM1 in Thornhill et al. (2021) is calculated with respect to preindustrial levels, whereas our all-sink value of 1.39 is appropriate for the present day. We opted to use the feedback factor of 1.49 (with respect to OH only) for the case described at L515 as we needed to determine the change in methane burden with change in OH. If a feedback factor of 1.39 is used with the equation now on line 517, we obtain a 3% difference between the value for dOH/dH2 obtained using this equation, and that obtained from Figure 1 (based on the box model methane LBCs in UKCA). We have amended the manuscript to reflect this uncertainty (see L559).

The choice of feedback factor influences our derived hydrogen GWP as it is used to determine the methane perturbation lifetime, used in the derived AGWP equations. We have amended our calculations to use the all-sink methane feedback factor of 1.39, which reduces the derived GWP by just under 0.3. However, this small change is offset by adjustments to the hydrogen lifetime in response to referee 1's comments. Table 3 has been amended to reflect both changes.

*L509. 2104 -> 2014.*

Changed.

*Matteo Bertagni*

*I am thrilled to see your recent publication on the hydrogen budget, a crucial topic for climate change mitigation scenarios. Congratulations on the valuable insights and research you have provided.*

*Our team recently published a related work about the methane feedback of hydrogen emissions, aiming to understand the potential impacts of the hydrogen economy on the methane budget (Bertagni et al., 2022, Nat. Comm). I noticed some similarities between our work and yours, for example, the transient methane growth due to an H2 perturbation (Fig. 5 in both papers) or the impact of H2 abundance on OH concentration (Fig. S2 in Bertagni et al.). It could be interesting to explore these similarities further or to discuss the differences between box- and climate-model results.*

Many thanks for the comments and pointing out some of the similarities between our results. To assess the performance of our equations, we had compared our derived transient methane response to box model simulations of an H2 pulse using the box model described in Section 2.2 and it is interesting to also see how the response compares to that modelled in Bertagni et al.. We have now added a sentence comparing the transient methane response determined using our equations with other model studies: 'A comparison of the transient methane growth due to an $H_2$ perturbation as determined by equations E1 to E3 in Figure 5 (grey line) with model studies where methane has been free-to-respond to an $H_2$ perturbation (including the box model described in Section 2.2 and the model studies of Derwent et al., 2020 and Bertagni et al., 2022) shows that the response represented by equations E1 to E3 is similar to that found in transient model simulations following an $H_2$ pulse emission.'

*One main difference I noticed is our perspectives on methane emission scenarios. While your paper suggests that H2 displacement of fossil fuels will decrease methane emissions, I believe this is true only for green H2 or blue H2 with very low methane leak rates. In contrast, blue H2 production with relatively classic methane leakage rates (2% or higher) could actually increase methane emissions (see Bertagni et al., or also Howarth and Jacobson 2021 for example). Perhaps this difference in perspective could be worth a comment in the manuscript, or it may be that the 'fourth set' of scenarios by the Authors only considers green H2.*

The fourth set of scenarios in our manuscript does assume that all hydrogen is generated using a green method and this should have been mentioned in the manuscript. A sentence has now been added to clarify this: 'In these scenarios, we assume that all hydrogen will be green hydrogen, with no emissions of other species associated with the $H_2$ production method. In the case that blue or grey hydrogen is used, it is possible methane emissions may increase due to emissions occurring during hydrogen production (e.g. Bertagni et al., 2022).'

---

## Author Response (AR2)

**Replies to Reviewers**

Again, we would like to thank the two referees for useful comments which have helped improve the manuscript. Our responses appear in black below.

**Reviewer 1**

*The authors have done a credible job of re-explaining and/or revising the manuscript in response to the extensive reviews. There is one point for which our viewpoints remain at right angles and one new curiosity that I found when reading the revised manuscript.*

*The authors' insistence on using air-mass weighted OH as a key variable is still fundamentally wrong and they know better. They know that the k(OH+CH4) and OH are unusually correlated and thus the product of the means as used here is clearly not equal to - and may not behave like - the mean of the product, which is related to the CH4 lifetime. At this stage, they do not want to redo all they fitting exercises and as a reviewer, I have done what I can. It should be published, but unfortunately, some of the analysis will not be as helpful or useful for future model comparisons.*

As appears in the manuscript, the methane production from H2, 'a' is calculated using the following equation:

a = CH4 * k(CH4+OH) * dOH/dH2

If we use OH weighted by CH4+OH rather than airmass to determine both k(CH4+OH) and dOH/dH2, we determine different values for k(CH4+OH) and dOH/dH2 individually, but the effects cancel and our value for 'a' remains unchanged. dOH/dH2 doubles and k(CH4+OH) halves. To aid future model comparisons, we now also quote values for k(CH4+OH) and dOH/dH2 where OH has weighted by CH4+OH, in addition to the values based on mass-weighted OH.

*The odd curiosity that I just noticed has to do with Figure 1D and discussion (~L240-245). Ignoring the CH4 feedback cases (dashed lines), one would expect that the tropospheric H2 injected at the tropopause will fall off like CH4 in the stratosphere and be converted to H2O by 50 km as observed (the lower bump in H2 being from CH4 production). This looks fine for the 750 ppb H2 case where the water increases as expected by (0.750 – 0.500) = 0.25 ppm! (light blue line). For +1000 ppb H2, however, the increase is half that expected (0.38 vs 0.50, orange line); and for the 2000 ppb H2 we barely get +1.2 ppm H2O instead of +2.0 ppm. What is going on with the sudden shift in H2O yield from H2? The authors might want to add a small comment on this.*

In our model simulations we do not get exactly 1 ppm of water vapour for every 1 ppm of H2 added to the model (we get less). However, we are not as far off this value as the reviewer suggests above as in the 2000 ppb H2 simulation, H2 has been increased by 1.5 ppm rather than 2.0 ppm. For this simulation, we get ~+1.2 ppm H2O for +1.5 ppm surface H2. We have added the following sentence to the manuscript 'Note that in our simulations

the maximum increase in stratospheric water vapour (occurring at ~55 km) is slightly less than 1 ppm per 1000 ppb increase in surface H2.'

**Reviewer 2**

*The revised paper is improved but I am still unhappy with some aspects:*

*l211-217 It is a bit odd to compare a methane lifetime with respect to OH with a total lifetime (i.e. with respect to OH, Cl, soil, and stratospheric sinks), given that it is straightforward to inter-convert between these two lifetimes, with simple assumptions about the Cl, soil and stratospheric lifetimes (e.g., see Prather et al., 2012; Stevenson et al., 2013). Please just straightforwardly compare like with like, and when referring to "the methane lifetime" (l217) be explicit about whether this is the total lifetime or with respect to OH. This is important as it is a major source of confusion in the literature, so all publications should be crystal clear about this to reduce this confusion.*

*l274 Close brackets*

Done.

*l300 CO2 emissions (I don't think we are anticipating reductions in CO2 concentrations anytime soon).*

Done.

*l343 The earliest publication with a GWP for H2 to my knowledge is Derwent et al (2001), so it would seem appropriate to reference that here.*

We have added a reference to Derwent et al. (2001) here.

*l470-477 The methane feedback factor is a strictly defined thing, and it uses the total methane lifetime with respect to all sinks. You can't just optionally define it differently! This is closely related to the first point above. Both reviewers queried the strangely high value for the methane feedback factor used - the authors have clarified why it was high (they calculated it using the incorrect methane lifetime) - they should just do this properly, and not seed confusion in the literature. It has only a minor impact on the final result.*

We have removed reference to more than one methane feedback factor for UKESM and only quote the value with respect to all sinks.

*Fix these points and I will be happy.*